Corrected: Author correction

# Sema7A/PlxnCl signaling triggers activity-dependent olfactory synapse formation

Nobuko Inoue[1], Hirofumi Nishizumi[1], Hiromi Naritsuka[2], Hiroshi Kiyonari[3] & Hitoshi Sakano[1]

In mammals, neural circuits are formed based on a genetic program and further refined by neuronal activity during the neonatal period. We report that in the mouse olfactory system, the glomerular map is not merely refined but newly connected to second-order neurons by odorant-receptor-derived neuronal activity. Here, we analyzed a pair of molecules, Sema7A, expressed in olfactory sensory neurons (OSNs) in an activity-dependent manner, and PlxnC1, localized to dendrites of mitral/tufted (M/T) cells in the first week after birth. In Sema7A or PlxnC1 knockout (KO) mice, initiation of synapse formation and dendrite selection of M/T cells were perturbed. Reconstitution and rescue experiments demonstrated that Sema7A–PlxnC1 interaction is essential to form the post-synaptic assembly. Pharmacological blocking experiments indicated that synaptic transmission triggers primary dendrite selection by synaptic competition. We conclude that Sema7A signaling is key to inducing activity-dependent post-synapse events and dendrite selection in M/T-cells during the neonatal period.

[1] Department of Brain Function, University of Fukui School of Medicine, 23-3 Shimo-aizuki, Matsuoka, Fukui 910-1193, Japan. [2] Department of Physiology, Graduate School of Medicine, The University of Tokyo, 7-3-1 Hongo, Bunkyo-ku, Tokyo 113-0033, Japan. [3] RIKEN Institute, 2-2-3 Minatojima-minamimachi, Chuo-ku, Kobe 650-0047, Japan. Correspondence and requests for materials should be addressed to H.S. (email: sakano.hts@gmail.com)

The mouse olfactory system detects a variety of odorants using a repertoire of ~1000 odorant receptors (ORs)[1]. Each olfactory sensory neuron (OSN) expresses only one functional OR gene[2], and OSN axons expressing the same OR species converge to a specific set of glomeruli in the olfactory bulb (OB)[3]. Since a given odorant activates multiple OR species[4], odor signals detected in the olfactory epithelium (OE) are converted to a unique combination of activated glomeruli, enabling the mammalian brain to discriminate a variety of odorants[5]. The odor information is further processed by local circuits in the OB and conveyed by mitral/tufted (M/T) cells for decision making to various areas in the olfactory cortex (OC)[6].

During embryonic development, a coarse olfactory map is generated by a combination of dorsal–ventral (D–V) patterning based on anatomical locations of OSNs, and anterior–posterior (A–P) patterning based on the OR-derived cAMP signals[7, 8]. After OSN axons reach their approximate destinations in the OB, further refinement of an olfactory map occurs through fasciculation of OSN axons in an activity-dependent manner[9]. Although most of these processes take place by axon–axon interactions of OSNs[10], proper connections are needed between OSN axons and the dendrites of M/T cells. In contrast to the primary projection of OSN axons, little is known about synapse formation with M/T cells. For example, what mediates the synapse formation between OSN axons and M/T-cell dendrites? How do they find their partners for proper synapse formation? What are the synaptic events involved in these processes? In an effort to address these questions, we searched for a pair of ligand and receptor molecules that are involved in synapse formation and dendrite selection of M/T cells.

## Results

**Sema7A expressed in OSN axons**. To determine what kinds of molecules are essential for initiating the synapse formation in glomeruli, we searched for a receptor and ligand pair each of which was expressed by OSN axons or M/T-cell dendrites. We examined 26 different genes of axon guidance, cell adhesion, and signaling molecules for their expression in the OE and OB (Table 1). Among them, Semaphorin 7A (Sema7A) and its candidate receptor, Plexin C1 (PlxnC1), were found to be expressed in OSN axons and M/T-cell dendrites, respectively. Immunostaining of OB sections revealed that Sema7A was localized to the axon termini of OSNs and distributed in a mosaic fashion in the glomerular map (Fig. 1a). Immuno-electronmicroscopy further demonstrated that Sema7A is localized to the pre-synaptic terminal of OSN axons (Fig. 1b). Levels of Sema7A expression are uniquely determined by OR species and vary among different glomeruli, e.g., high in rI7, medium in MOR29B, and low in MOR29A glomeruli (Fig. 1c). This observation indicates that Sema7A expression is downstream of the OR activity as reported for glomerular segregation molecules, e.g., Kirrel2 and Kirrel3[9].

To examine the activity dependency of Sema7A expression, we performed uni-lateral naris occlusion at postnatal day 0–6 (P0-6). When one naris was occluded, Sema7A expression in the OE was markedly reduced on the occluded side (Fig. 1d left), showing that Sema7A expression is regulated by stimulus-driven OSN activity. Activity dependency of Sema7A expression was confirmed by the analysis of mutant mice lacking CNG-A2, a component of the cyclic nucleotide-gated (CNG) channel[11]. Since the *CNG-A2* gene is carried on the X chromosome, mosaicism can occur among OSNs for CNG-A2 activity due to stochastic X-chromosome inactivation in the female[12]. In the CNG-A2$^{+/-}$ mice, duplicated glomeruli are formed: one is for CNG-A2$^+$ axons and the other for CNG-A2$^-$. When rI7 glomeruli were analyzed at P5, Sema7A levels in the CNG-A2$^-$ glomerulus were much lower than in that of CNG-A2$^+$ (Fig. 1d middle). We also analyzed the mutant rI7 (RDY) defective in G-protein coupling[8] for its ability to produce Sema7A. In OSNs where rI7 (RDY) was expressed, Sema7A

---

**Table 1 Expression of axon guidance, signaling, and adhesion molecules in the OE and OB**

| Gene | Name | Function | OE (mRNA) | OB (mRNA) | M/T dendrites (Protein) |
|---|---|---|---|---|---|
| *ephA5* | Ephrin type-A receptor 5 | Protein-tyrosine kinase | + | − | − |
| *efna5* | ephrin-A5 | Ligand of Eph-related Kinase | + | + | − |
| *s100A5* | S100 calcium binding protein A5 | Calcium binding | + | − | − |
| *kirrel2* | kin of IRRE like 2 | Cell adhesion | + | − | − |
| *kirrel3* | kin of IRRE like 3 | Cell adhesion | + | + | − |
| *pcp4* | Purkinje cell protein 4 | Calmodulin binding | + | − | − |
| *ptprn* | protein tyrosine phosphatase type-N receptor | Signal transduction | + | + | − |
| *ptprf* | protein tyrosine phosphatase type-F receptor | Signal transduction | + | − | − |
| *nrp1* | neuropilin 1 | Signal transduction | + | + | − |
| *nrp2* | neuropilin 2 | Signal transduction | + | + | − |
| *nph3* | neurexophilin 3 | Neuropeptide | + | − | − |
| *sema3a* | semaphorin 3a | Axon guidance | + | + | − |
| *sema3f* | semaphorin 3f | Axon guidance | + | − | − |
| *sema4d* | semaphorin 4d | Axon guidance | − | + | − |
| *sema7a* | semaphorin 7a | Axon guidance | + | + | − |
| *plxna1* | plexin a1 | Signal transduction | + | − | − |
| *plxna2* | plexin a2 | Signal transduction | − | − | − |
| *plxna3* | plexin a3 | Signal transduction | + | + | − |
| *plxna4* | plexin a4 | Signal transduction | − | − | − |
| *plxnb1* | plexin b1 | Signal transduction | + | − | − |
| *plxnb2* | plexin b2 | Signal transduction | + | − | − |
| *plxnb3* | plexin b3 | Signal transduction | + | − | − |
| *plxnc1* | plexin c1 | Signal transduction | − | + | + |
| *ingb1* | integrin beta 1 | Cell adhesion | − | − | − |
| *omp* | olfactory marker protein | Signal transduction | + | − | − |
| *pcdh21* | protocadherin-21 | Cell adhesion | − | + | + |

OE and OB sections at P4 were analyzed by in situ hybridizaion, OB sections were analyzed by immunohistochemistry. +, detected; −, not detected

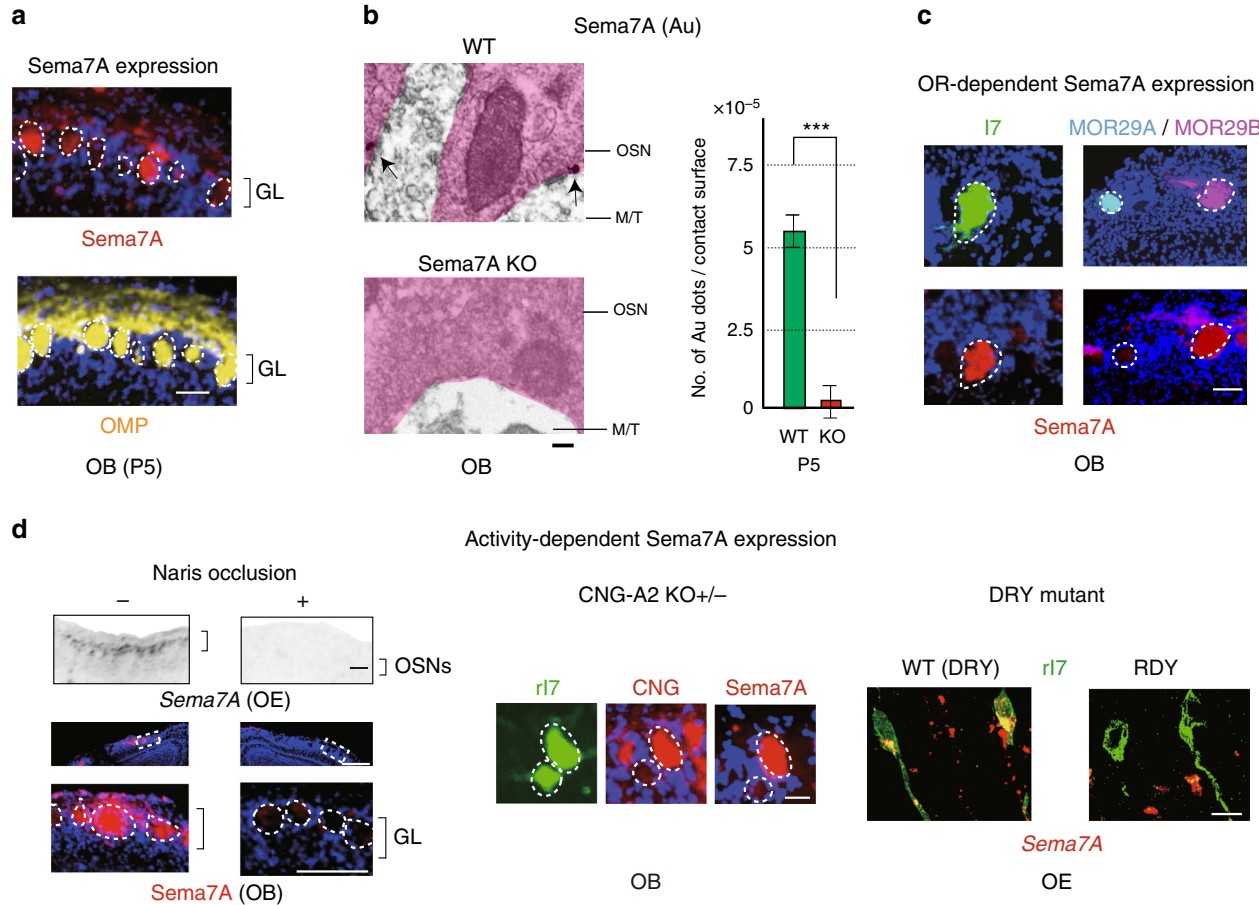

**Fig. 1** Activity-dependent Sema7A expression in OSN axons. **a** Distribution of Sema7A in the glomerular layer of the OB. OB sections of the WT at P5 were immunostained with anti-Sema7A and anti-OMP antibodies. Scale bar=25 μm. **b** Detection of Sema7A in the axon termini of OSNs. Immuno-electronmicroscopy identified Au-labeled Sema7A (arrows) in the pre-synaptic termini of OSN axons at P5. No Au signals were detected in the Sema7A KO. The number of signals per contact surface was counted (right). ***$p < 0.005$ (Student's $t$-test). Error bars indicate SD. $n = 3$ animals for each. Scale bar=300 nm. **c** Variable expression of Sema7A among different glomeruli. OB sections at P10 were immunostained with anti-Sema7A antibodies. EYFP-tagged rI7, ECFP-tagged MOR29A and EYFP-tagged MOR29B glomeruli were detected by immunostaining with anti-GFP antibodies. Scale bar=25 μm. **d** Activity-dependent Sema7A expression. Left, Down-regulation of Sema7A expression by uni-lateral naris occlusion. Mice were unilaterally occluded at P0. OE sections (P6) were analyzed by in situ hybridization. OB sections (P6) were analyzed by immunostaining with anti-Sema7A antibodies. (+) occluded, (−) unoccluded. Scale bars, 50 (upper), 100 (lower) μm. Middle, CNG-channel-dependent Sema7A expression. Duplicated glomeruli of rI7 in the heterozygous CNG-A2 KO were analyzed for Sema7A expression at P5. EYFP-tagged rI7 glomeruli were detected by immunostaining with anti-GFP antibodies. OB sections were immunostained with antibodies against CNG-A2 and Sema7A. Scale bar=25 μm. Right, OR-activity dependent Sema7A expression. A DRY-motif mutant of rI7 suppresses *Sema7A* expression because it does not generate cAMP. OE sections expressing the WT rI7 and DRY-motif mutant (RDY) were analyzed by in situ hybridization for *Sema7A* expression at P5. OSNs expressing the EYFP-tagged rI7 were detected by immunostaining with anti-GFP antibodies. Scale bar=10 μm. GL, glomerular layer

expression was diminished (Fig. 1d right), because cAMP responsible for opening CNG-A2 channels is not produced by the mutant rI7. These results demonstrate that Sema7A expression is regulated by OR-derived neuronal activity. Since all kinds of ORs generate minimum levels of Sema7A in neonates, this activity could be the spontaneous OR activity reported by Reisert[13].

**PlxnC1 expressed in M/T cell dendrites**. We next studied the candidate receptors for Sema7A in M/T cells. Both in the immune and in the central nervous systems, PlxnC1 and Integrin β1 are known to serve as receptors for Sema7A[14–16]. We, therefore, examined whether these receptors are expressed in M/T cells by in situ hybridization (ISH) of OB sections. At early developmental stages, *PlxnC1*, but not *Integrin β1*, is transcribed in M/T cells (Fig. 2a). PlxnC1 protein is localized to M/T-cell dendrites

(Fig. 2b). Staining of PlxnC1 is weak and patchy in glomeruli in the newborn, becomes stronger at P3–P5, and gradually disappears after P8. Contrary to Sema7A, PlxnC1 is detected evenly in all glomeruli at P3–5 regardless of OR species. In the adult, both PlxnC1 and Integrin β1 are produced in M/T cells, but neither of them is detected in M/T-cell dendrites within glomeruli (Fig. 2b).

We then examined whether Sema7A in OSN axons indeed interacts with PlxnC1 in M/T-cell dendrites by alkaline phosphatase (AP) staining[9, 17]. PlxnC1 and Sema7A were each fused to AP, and resulting fusion proteins were used as affinity probes. Serial OB sections were treated with AP-fusion proteins and developed with AP substrate. OB sections were also immunostained with antibodies against Sema7A or PlxnC1. AP-fusion proteins of PlxnC1 and Sema7A detected the same glomerular patterns as anti-Sema7A and anti-PlxnC1 antibodies did, respectively (Fig. 3). Each KO was analyzed as a negative

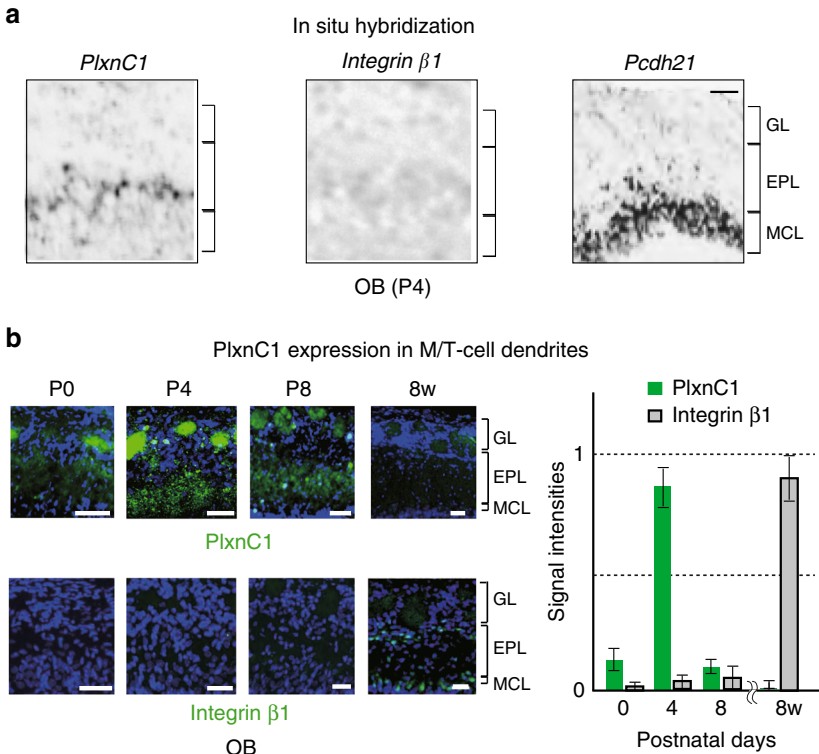

**Fig. 2** PlxnC1 localized to the M/T-cell dendrites as Sema7A receptor in neonates. **a** Expression of candidate receptors for Sema7A in the OB. OB sections at P4 were analyzed by in situ hybridization using the probes for *PlxnC1*, *Integrinβ1*, and *Pcdh21* (M/T-cell marker). *n* = 6 animals. Scale bar=30 μm. **b** Temporal regulation of PlxnC1 expression in the OB. OB sections were immunostained with antibodies against PlxnC1 and Integrinβ1 at various developmental stages. Relative signal intensities (GL/EPL) are compared at different time points. *n* = 2 animals except for P4 (*n* = 6). Scale bars=30 μm. GL glomerular layer, EPL external plexiform layer, MCL mitral cell layer

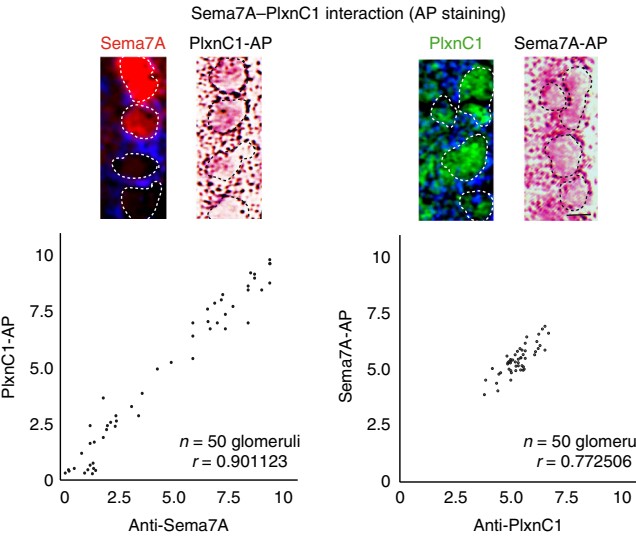

**Fig. 3** Physical interaction of Sema7A and PlxnC1 in the OB. Sema7A–PlxnC1 interaction was detected by the binding assay with AP-fusion proteins. Two consecutive OB sections at P4 were stained with PlxnC1-AP and immunostained with antibodies against Sema7A (left). OB sections were also stained with Sema7A-AP and immunostained with antibodies against PlxnC1 (right). Positive correlations are found between the AP-staining and immunostaining signals. The Pearson's correlation coefficient was calculated (*r*). *n* = 50 glomeruli. Scale bar=25 μm

control. These observations indicate that PlxnC1 serves as a receptor for Sema7A in its interaction between OSN axons and M/T-cell dendrites.

**KO studies of Sema7A/PlxnC1 signaling.** To examine the in vivo function of Sema7A signaling, we analyzed the Sema7A KO[14] after crossing with Tg MOR29B[18] or MOR28 knock-in mice[19]. Since the KOs of other semaphorin molecules, e.g., Sema3A and Sema3F, severely affect OSN projection to the OB[7, 8], we examined glomerular map formation in the Sema7A KO. Whole mount staining of the OB from the Sema7A KO at P8 did not reveal any notable defects in OSN projection or in glomerular formation (Supplementary Fig. 1 and Supplementary Table 1). However, synapse formation and dendrite maturation appeared to be perturbed in the Sema7A KO. Immunostaining of rI7 glomeruli with synaptic markers, vGlut2 and GluR1[20–22], revealed that synapse formation was impaired in the Sema7A KO at P5 (Fig. 4a). It is notable that synapse formation remained defective in the Sema7A KO even into adulthood (Fig. 4a). We then studied dendrite selection of M/T cells for the rI7 glomeruli in the Sema7A KO. To visualize dendrites[23], Lucifer yellow (LY) was injected into the glomerular structure for EYFP-tagged rI7 and connecting M/T-Cell dendrites were analyzed by immunostaining with anti-GFP and LY antibodies. We counted the numbers of M/T-cells with primary dendrite as mature and with multiple dendrites as immature. In the Sema7A KO, dendrite selection of M/T cells was perturbed, although dendrite extension to the glomerular layer (GL) was normal (Fig. 4b).

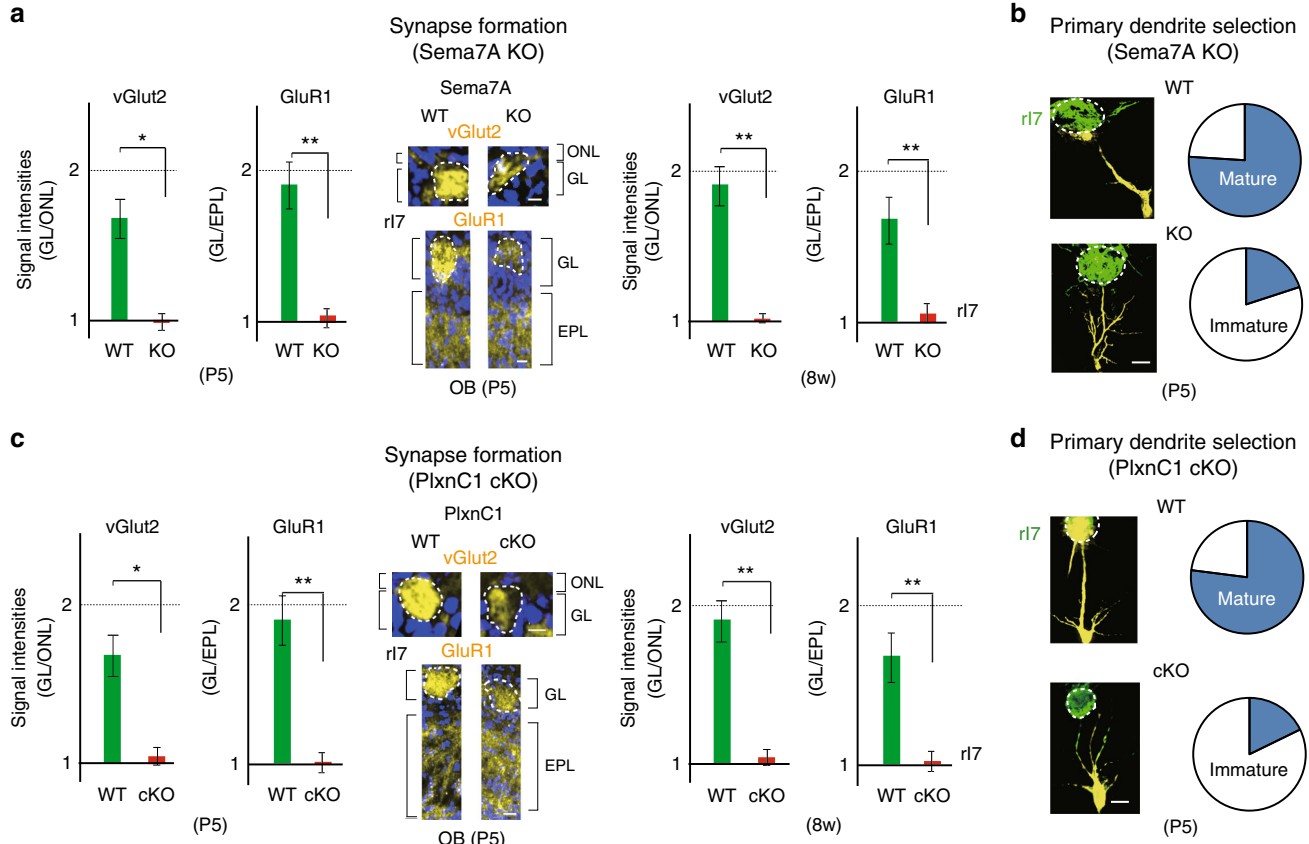

**Fig. 4** Synapse formation and dendrite maturation in the Sema7A and PlxnC1 KOs. **a** Synapse markers within glomeruli in the Sema7A KO. OB sections of the WT and Sema7A KO at P5 and 8w were immunostained with antibodies against vGlut2 and GluR1, pre- and post-synaptic markers, respectively. Staining photos are shown for P5 sections. Signal intensities within the rI7 glomeruli were normalized by the pre-synaptic signals in the ONL and post synaptic signals in the EPL. Fluorescent signals of GL/ONL and GL/EPL in glomeruli are compared between the WT and Sema7A KO. Scale bars=25 μm. *$p$ < 0.05, **$p$ < 0.01 (Student's $t$-test). Error bars indicate SD ($n$ = 5, 6, 2, 3 animals). **b** Dendrite morphology of M/T cells in the Sema7A KO. EYFP-tagged rI7 glomeruli (green) were detected by immunostaining with anti-GFP antibodies. M/T cells (yellow) were visualized by Lucifer yellow (LY) injection into the rI7 glomeruli of the WT and Sema7A KO at P5 (left). The numbers of M/T cells with one dendrite (mature) and those with multiple dendrites (immature) to GL were counted for the rI7 glomeruli. The ratios (%) of mature (blue) and immature (white) M/T cells are shown (right): WT, 26/34 (76.5 %); KO, 8/34 cells (23.5 %). $n$ = 20 glomeruli. Scale bar=10 μm. **c** Synapse markers within glomeruli in the PlxnC1 cKO. OB sections of the WT and PlxnC1 cKO at P5 and 8w were analyzed for synapse markers. Staining photos are shown for P5 sections. Signal intensities within the rI7 glomeruli were shown. Scale bars=25 μm. *$p$ < 0.05, **$p$ < 0.01 (Student's $t$-test). Error bars indicate SD ($n$ = 5, 5, 2, 3 animals). **d** Dendrite morphology in the PlxnC1 cKO. M/T cells (yellow) were visualized by LY injection into the rI7 glomeruli of the WT and PlxnC1 cKO at P5 (left). The ratios (%) of mature (blue) and immature (white) M/T cells are shown (right): WT, 24/31 (77.4 %); KO, 7/31 (22.6 %). $n$ = 23 glomeruli. Scale bar=20 μm

We also analyzed the PlxnC1 KO mice. The M/T-cell-specific conditional KO (cKO) was generated for PlxnC1 using the Pcdh21-Cre mouse[24] as a driver. It was confirmed that PlxnC1 was absent in the OB of the KO by immunostaining, although the MCL was normally formed (Supplementary Fig. 2). We studied synapse formation and dendrite selection of M/T cells. In the cKO, M/T-cell layers were normally formed and dendrite extension took place. However, synapse formation was impaired (Fig. 4c) and dendrite selection was significantly delayed (Fig. 4d) as seen in the Sema7A KO. Synaptic structures were also studied by electron microscopy (EM) in the Sema7A KO and PlxnC1 cKO (Fig. 5a, b). It was confirmed that initiation of synapse formation was blocked and post-synaptic density (PSD) was not formed, although membrane attachment was taking place normally. Compared with the WT, synaptic markers were not detected in the PlxnC1 cKO even in the adult (Fig. 4c). These results suggest that PlxnC1 is needed for synapse formation and dendrite selection of M/T cells.

It is interesting that in the Sema7A[high]-type glomeruli (e.g., for MOR29B), synapse formation and dendrite selection occur a few days earlier than in the Sema7A[low]-type (e.g., for MOR29A). This was shown by immunostaining of synaptic markers and LY staining of M/T-cell dendrites at P4 (Fig. 6). These observations suggest that Sema7A expression promotes synapse formation and dendrite selection of M/T cells in neonates.

**Sema7A–PlxnC1 interaction induces post-synaptic events.** In addition to in vivo analyses with KOs, we also studied how the Sema7A–PlxnC1 interaction induces synapse formation using the heterologous system. We introduced the secrete-type of *Sema7A* and *myc*-tagged *PlxnC1* genes into HEK293 cells in culture (Fig. 7a). In this system, the secreted Sema7A interacts with PlxnC1 expressed on the HEK-cell surface (Fig. 7b). To analyze the aggregation of PSD, cells had been transfected with the *FLAG*-tagged gene for SAP90 that serves as a scaffold molecule[25] in clustering the PSD. PSD formation is known to be the earliest hallmark of post-synaptic events in M/T-cell dendrites[26, 27]. Staining signals of FLAG (SAP90) was found to be co-localized with those of myc (PlxnC1) when the full-length PlxnC1 was

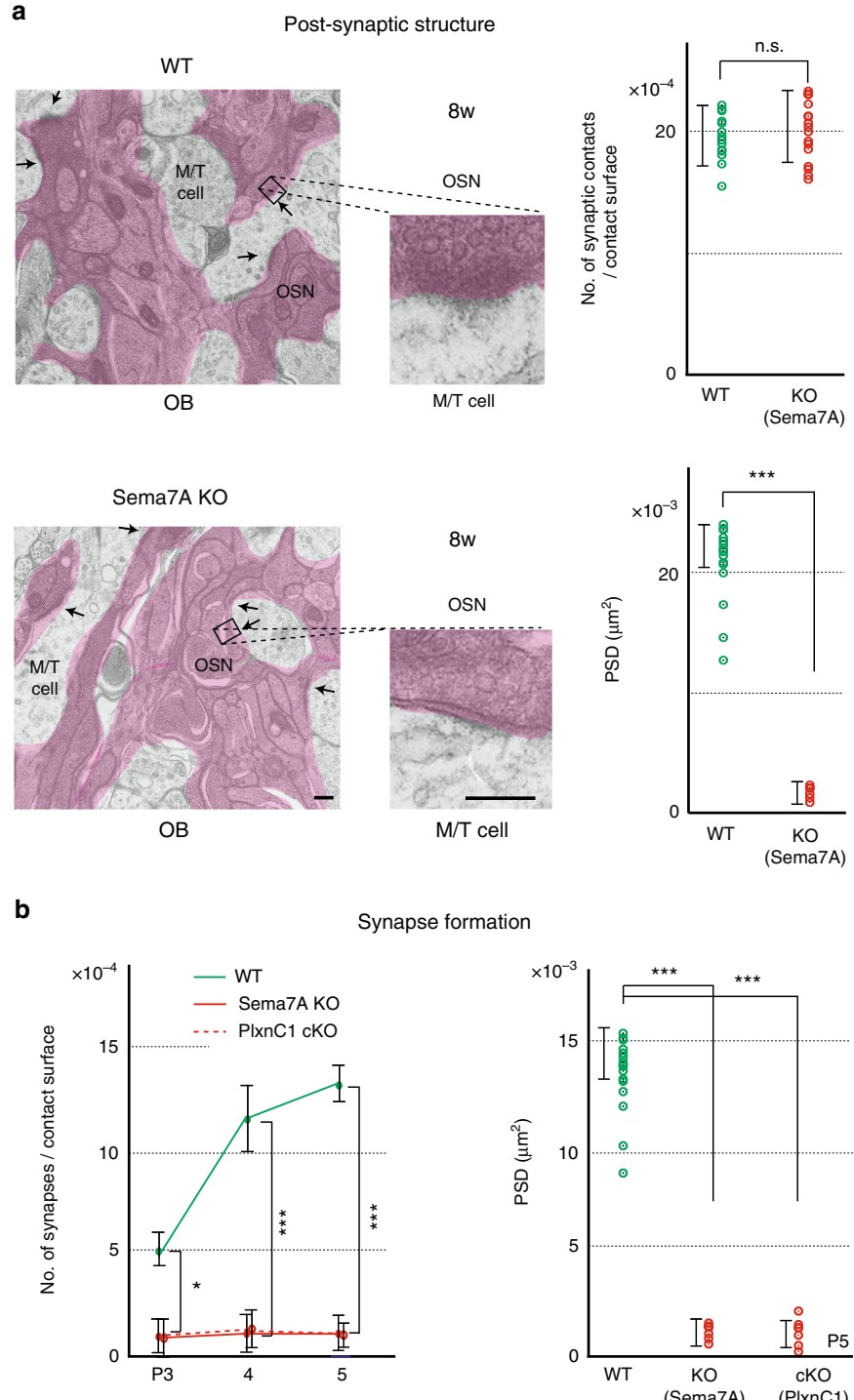

**Fig. 5** Synapse formation in the KOs of Sema7A/PlxnC1 signaling. **a** EM analysis of post-synaptic structures. Synaptic contacts and post-synaptic structures were analyzed in the 8w WT, and Sema7A total KO by electron microscopy (EM). Arrows indicate synaptic contacts. Enlarged views demonstrate that only the surface attachment, but not mature PSD, is formed in the KO. ***$p < 0.005$ (Student's $t$-test). n.s., not significant. Error bars indicate SD ($n = 3$, 4 animals). Scale bars=300 nm. **b** The numbers of synaptic structures are compared among the WT, Sema7A total KO, and PlxnC1 cKO in neonatal stages by electron microscopy (left). *$p < 0.05$, ***$p < 0.005$ (Student's $t$-test). Error bars indicate SD ($n = 2$, 2, 5 animals for each line). The areas of PSD at P5 are also compared (right). ***$p < 0.005$ (Student's $t$-test). Error bars indicate SD ($n = 6$, 5, 6 animals)

expressed with the secrete type of Sema7A in HEK cells (Fig. 7b, c).

Based on the 3D-structural analyses of Sema7A[28], we introduced amino acid substitution, Y → S at residue 213, into the interaction site of Sema7A with PlxnC1. Interestingly, co-localization of SAP90 with PlxnC1 was blocked with the mutant Sema7A, Y213S (Fig. 7b, c). Co-localization was also blocked when the cytosolic domain of PlxnC1 was deleted (ΔC PlxnC1) (Fig. 7b, c). In these experiments, the full-length and ΔC PlxnC1 molecules were separately detected with antibodies against N- and C-terminal regions of PlxnC1, respectively. Binding of the secreted Sema7A to the cell surface was confirmed by

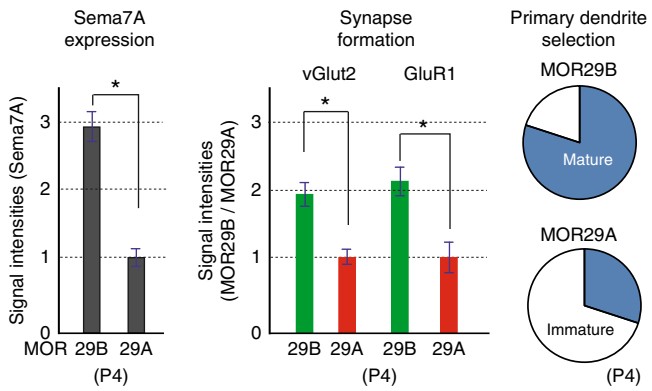

**Fig. 6** Sema7A promotes dendrite maturation and synapse formation. Left, Sema7A levels in the MOR29B and MOR29A glomeruli. OB sections at P4 were immunostained with anti-Sema7A antibodies, and signal intensities of Sema7A are compared (see also Fig. 1c). *$p < 0.05$ (Student's $t$-test). Error bars indicate standard deviation SD ($n = 8$ glomeruli). Middle, Pre- and post-synaptic markers in the MOR29A and MOR29B glomeruli. OB sections at P4 were also immunostained with antibodies against vGlut2 and GluR1, pre- and post-synaptic markers, respectively. Relative fluorescent signals (MOR29B/MOR29A glomeruli) are compared. *$p < 0.05$ (Student's $t$-test). Error bars indicate standard deviation SD ($n = 8$ glomeruli). Right, Dendrite maturation of M/T cells in the MOR29A and MOR29B glomeruli. M/T-cell dendrites were visualized by LY injection into the MOR29A and MOR29B glomeruli at P4. The ratios (%) of mature and immature M/T cells in the MOR29A and MOR29B glomeruli are shown: MOR29A, 8/29 (27.6 %); MOR29B, 22/28 (78.6%). $n = 16$ glomeruli

immunostaining (Fig. 7b). These results suggest that signaling of Sema7A–PlxnC1 interaction recruits SAP90 to initiate the post-synaptic events in M/T cells.

**Signaling pathways for the synaptogenesis in M/T cells.** Small GTPases play key roles in Plxn-mediated signaling and synaptic functions[29]. It has been reported that both Rac1/Cdc42 and Ras pathways are downstream of PlxnA or PlxnB signaling[29] and promote synapse formation[30]. Since Rac1/Cdc42 and Ras signals phosphorylate PAK and ERK[29], respectively, we analyzed rI7 glomeruli for PAK and ERK phosphorylation in the PlxnC1 cKO at P3. Immunostaining revealed that phosphorylation of PAK, but not ERK, was markedly reduced in the PlxnC1 cKO (Fig. 8a). The same results were also obtained with the Sema7A KO (Fig. 8a). It should be noted that the amount of proteins, as well as the transcripts of PAK and ERK, were not changed in the KOs. We then introduced the transgenic (Tg) Sema7A gene with the activity-independent MOR23 promoter[9] into the Sema7A KO background. Interestingly, constitutive expression of the Tg Sema7A, but not the interaction mutant Sema7A (Y213S), rescued the phosphorylation of PAK, but not of ERK, in the Sema7A KO mice (Fig. 8b). These results suggest that Sema7A–PlxnC1 interaction activates the Rac1/Cdc42 pathway, but not the Ras, to induce post-synaptic events in M/T-cell dendrites (Fig. 8d). This is the first demonstration that signaling via PlxnC1 regulates the Rac1/Cdc42-PAK pathway.

As mentioned earlier, not only synapse formation but also dendrite selection is perturbed in the Sema7A KO and PlxnC1 cKO, demonstrating that synaptic activity is needed for dendrite selection and branch pruning. We postulate that synaptogenesis triggered by Sema7A–PlxnC1 signaling induces an increase in calcium concentrations in M/T-cell dendrites. To examine this, we used a NMDA inhibitor, MK801, to pharmacologically block $Ca^{2+}$ entry into M/T cells[31] (Fig. 8c). This treatment blocks

synaptic activity after the synapses have been formed. Transcription levels of cpg15, a marker gene for synaptic activity[32], were measured in M/T cells with or without MK801 treatment. As expected, MK801 treatment markedly reduced cpg15 expression (Fig. 8c left). When the rI7 glomeruli were examined, dendrite selection was suppressed by MK801 (Fig. 7c right). These results suggest that synaptic transmission triggers primary dendrite selection in M/T cells by synaptic competition.

## Discussion

In the mouse olfactory system, Sema and Plxn molecules have been extensively studied for their roles in targeting OSN axons, where both are produced in the OE for axon-axon interactions of OSNs[7, 8]. Although the primary projection to the OB is well-understood, little is known about synapse formation with second-order neurons, M/T cells. Our present study revealed that interactions between Sema7A expressed in OSN axons and PlxnC1 in M/T-cell dendrites are essential for triggering the activity-dependent post-synaptic events. Since other Sema genes in the olfactory system, e.g., Sema3A and Sema3F, are known to regulate targeting of OSN axons independent of neuronal activity[7, 8], Sema7A is an unique example of the Sema family genes that are involved in regulating synapse formation. In Purkinje cells, Sema7A was reported to facilitate synapse elimination of mossy fibers via PlxnC1in an activity-dependent manner[33]. However, expression profiles of Sema7A and PlxnC1 (pre vs. post synapses) and the effect of Sema7A–PlxnC1 signaling (synapse maturation vs. elimination) are opposite to what is found in the olfactory system. It will be interesting to study how Sema7A/PlxnC1 signaling functions in the two synaptic systems in different ways.

In the mouse olfactory system, interaction of Sema7A with PlxnC1 triggers post-synaptic assembly. How does Sema7A signaling recruit PSD molecules? One possibility is that binding of Sema7A to PlxnC1 induces the conformational change in PlxnC1[28], which may allow the recruitment and assembly of PSD molecules. Another possibility is that Sema7A–PlxnC1 interactions activate signal transduction pathways. Recent studies revealed that some transmembrane molecules initiate PSD formation using their own kinase activity, and not by recruiting PSD components via PSD domains[34]. Our study is the first to demonstrate that Sema7A signaling activates the Rac1/Cdc42-PAKpathway that regulates vesicle trafficking, cytoskeletal dynamics and gene expression necessary for the post-synaptogenesis in M/T cells.

It has been reported that dendrite selection and synapse formation of M/T cells are significantly delayed but take place in the CNG-A2 KO at later stages of development[11]. In the CNG-A2 KO, other ion-channels in OSNs may compensate the CNG-derived activity and induce basal expression of Sema7A for triggering the post-synaptic events in M/T cells (Supplementary Fig. 3). Like in the CNG-A2 KO, dendrite maturation of M/T cells is delayed but ultimately occurs in the Sema7A KO and PlxnC1 cKO. Interestingly, however, synapse formation remains perturbed in the latter two KOs even in the adult. Although these issues have to be clarified in the future, our result serves as a good example of the synaptotropic hypothesis formulated by Vaughn[35], in which synaptic activity is assumed to have a critical role in stabilizing the dendrite structure.

Sema7A levels are different from glomerulus to glomerulus and uniquely determined by the expressed OR species, thus showing mosaic distribution of Sema7A in the glomerular map. Is there any functional significance for this differential expression and mosaic distribution of Sema7A in the OB? As found in the present study, dendrite selection occurs at different times, depending

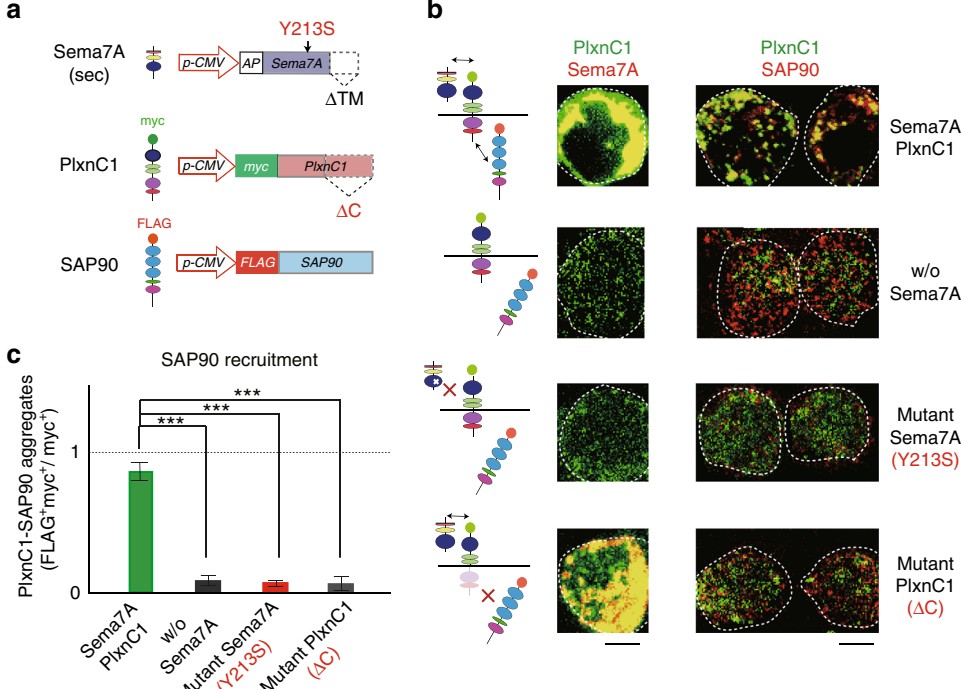

**Fig. 7** Reconstitution of Sema7A–PlxnC1 interaction in the HEK cells. **a** Schematic diagrams of vector constructs. Plasmid structures are illustrated for the secreted form of *Sema7A*, *myc*-tagged *PlxnC1*, and *FLAG*-tagged *SAP90*. Y213S mutation in Sema7A and ΔC deletion in PlxnC1 are indicated. **b** Co-expression of Sema7A, PlxnC1 and SAP90 in HEK cells. Secrete-form *Sema7A*, *myc*-tagged *PlxnC1* and *FLAG*-tagged *SAP90* genes were introduced into the HEK293 cells in culture. Structures of Sema7A, PlxnC1, and SAP90 are schematically shown (left). Cell-surface expression of Sema7A and PlxnC1 was separately analyzed by immunostaining (middle). To detect SAP90 and PlxnC1, cells were immunostained with antibodies against epitope tags FLAG and myc, respectively (right). When the secrete-form Sema7A was co-expressed, clustering of the PSD was induced. Co-expression of mutant Sema7A, Y213S, blocked PSD formation. PSD formation was also blocked with the mutant PlxnC1, ΔC. HEK cells are circled. Scale bars=10 μm. **c** Formation of SAP90 aggregates with PlxnC1. Yellow-stained aggregate-signals (PlxnC1 bound SAP90) are compared among the four different experiments, shown in **b**. Aggregate formation was calculated as myc$^+$FLAG$^+$ dots (yellow)/all myc$^+$ dots (green and yellow). $^{***}p < 0.005$ (Student's *t*-test). Error bars indicate SD (*n* = 3 trials)

on the expressed OR species. Since OR-derived neuronal activities in OSNs correlate with expression levels of Sema7A in glomeruli, dendrite selection proceeds faster for glomeruli that express higher-levels of Sema7A than those expressing lower-levels. In addition, OSN projection and glomerular maturation occur at different times for different OR species, as the onset of *OR* genes varies during development[36]. Differential synapse formation based on the expression levels of Sema7A, may lead to the ordered segregation of cortical projection during development.

ISH analysis revealed that both *PlxnC1* and *Integrin β1* are transcribed in M/T cells. However, neither PlxnC1 nor Integrin β1 protein is detected in M/T-cell dendrites in the adult. Moreover, Sema7A is cleaved by Caspase-9 in the aged animal[37]. Therefore, Sema7A signaling may not be needed for synapse formation in the adult. In the mouse olfactory system, OSNs are constantly regenerated to replace old axons with new ones[5]. In contrast, M/T cells are not replaced and glomerular structures are kept stably throughout the life[38]. In this connection, in the neuromuscular junction, the post-synaptic structure is maintained without the pre-synapse[39], and interneurons help the post-synaptic structures to survive[40]. It appears that Sema7A signaling is needed only for the formation of post-synaptic structure within glomeruli at the neonatal stage. Then, how does pre-synaptic maturation occur in OSN axons? Since GPI-anchored Sema7A in OSNs has no cytosolic domain, it is unlikely that Sema7A signaling directly induces pre-synaptic maturation. We postulate that pre-synaptic structure can be generated even in the absence Sema7A signaling, once the post-synaptic structure is established.

It is possible that post-synaptic PSD-maturation induces pre-synaptic formation in a retrograde manner by using other membrane proteins or membrane-permeable factors.

In the mouse olfactory system, global targeting of OSNs occurs first independent of neuronal activity, followed by activity-dependent local refinement[7–10]. Similarly, synapse formation of M/T cells appears to take place in a stepwise fashion; global alignment occurs first with nearby glomeruli, which is followed by the activity-dependent selection of M/T cell dendrites. In *Drosophila*, projection neurons are pre-specified by cell lineage and birth order to form synapses with incoming axons of ORNs[41]. This genetically programmed pre-specification of ORNs generates hard-wired circuits in the fly, which can induce stereotyped odor responses. In contrast, in the mouse olfactory system, synapse formation within glomeruli is modulated by OR-derived OSN activity. It will be interesting to determine whether the hard-wired circuit formation takes place normally when Sema7A signaling is blocked. Since innate odor responses are genetically programmed, they may not be affected in the KO of Sema7A or PlxnC1.

Considering that Sema7A expression is activity dependent and PlxnC1 localization in dendrites is restricted to the first week after birth, Sema7A signaling may play an important role in establishing proper olfactory circuits during the neonatal period. This may be accomplished by selecting dendrites and axon branches of M/T cells in an activity-dependent manner. Furthermore, environmental inputs in neonates may also activate a particular set of M/T cells leading to the formation of new neural circuits that

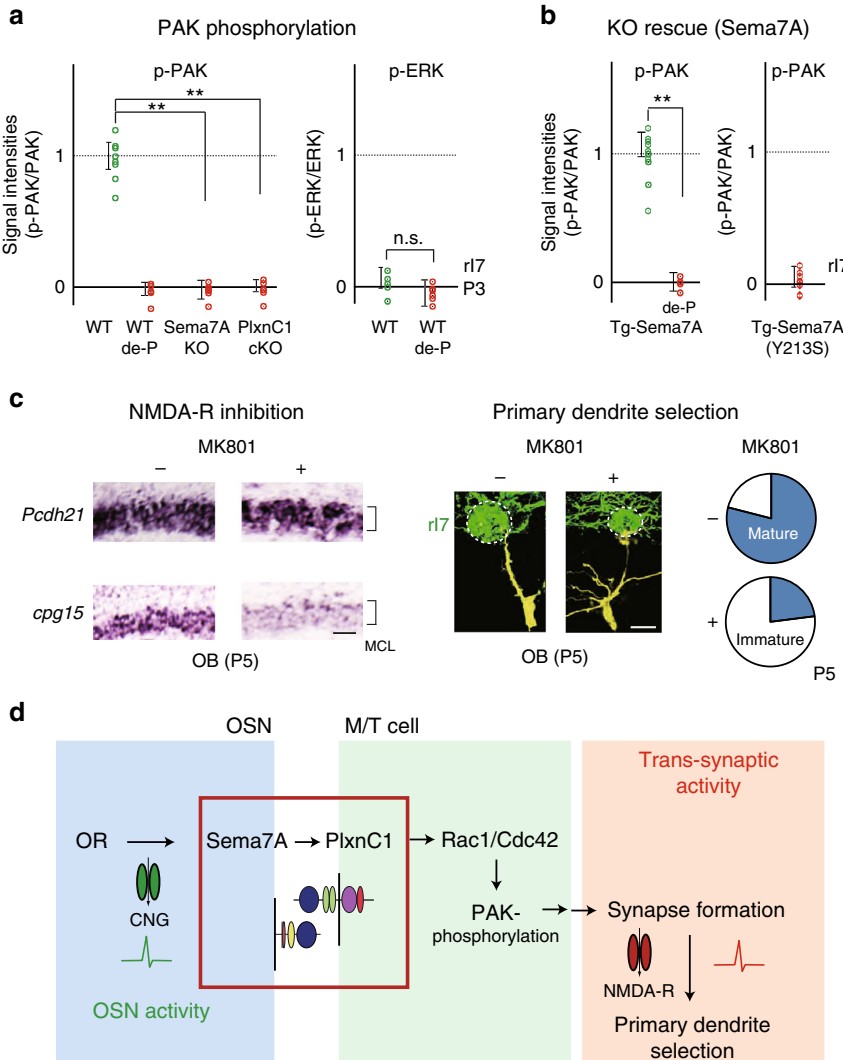

**Fig. 8** Downstream of Sema7A/PlxnC1 signaling. **a** Signal cascades downstream of Sema7A–PlxnC1 interaction. Left, Phosphorylation of PAK and ERK in the Sema7A KO and PlxnC1 cKO. OB sections at P3 were immunostained with antibodies against phosphorylated form, and total proteins. Phosphatase-treated samples (de-P) were also analyzed. Relative fluorescent intensities, p-PAK/PAK and p-ERK/ERK, are compared in the rI7 glomeruli. $^{**}p < 0.01$ (Student's *t*-test). Error bars indicate SD ($n = 7, 3, 4, 5$ glomeruli for WT, de-P, Sema7A KO, and PlxnC1 cKO). **b** Recovery of PAK- phosphorylation in the Sema7A KO by introducing the Tg *Sema7A* gene. This rescue is blocked by the Sema7A mutation, Y213S, which interferes with PlxnC1 interaction. By mating the transgenic animal, the WT or mutant *Sema7A* gene was introduced into the Sema7A KO mice, and was expressed constitutively in the rI7-expressing OSNs. OB sections at P3 were immunostained with antibodies against phosphorylated and total PAK. Phosphatase-treated samples (de-P) were also analyzed. Relative fluorescent intensities p-PAK/PAK in the rI7 glomeruli are compared between the WT and mutant Sema7As. $^{**}p < 0.01$ (Student's *t*-test). Error bars indicate SD ($n = 3$ mice for the WT Sema7A, $n = 2$ mice for the mutant Sema7A). **c** Inhibition of the trans-synaptic activity in M/T cells. Left, Inhibition of the trans-synaptic activity with an NMDA-R inhibitor, MK801. OB sections at P5 were analyzed by in situ hybridization for the expression of *Pcdh21* and *cpg15*. EYFP-tagged rI7 glomeruli were identified by immunostaining. The expression of *cpg15* (synaptic-activity marker) in M/T cells was inhibited after MK801 injection. $n = 3$ animals. Scale bar = 20 μm. Right, Dendrite morphology of the MK801-treated M/T cells at P5. M/T cells were stained in yellow by LY injection into the rI7 glomeruli. The ratios (%) of mature (blue) and immature M/T cells in the rI7 glomeruli are shown: MK801⁻, 19/25 (76.0 %); MK801⁺, 6/25 (24.0 %). $n = 19$ glomeruli. Scale bar = 10 μm. **d** Schematic diagram of synapse formation regulated by Sema7A signaling. OR-derived-neuronal activity induces Sema7A expression in OSNs. Sema7A interacts with PlxnC1 expressed in M/T-cell dendrites and activates the Rac1/Cdc42-PAK signaling cascade to induce synapse formation. Trans-synaptic activities in M/T cells are responsible for dendrite selection and maturation. MCL, mitral-cell layer

mediate adaptive olfactory responses. These intriguing possibilities have to be examined in the future. The mouse olfactory system will continue to serve as an excellent system to study the sensory circuit formation and modulation of decision making in the mammalian brain.

## Methods

**Mice**. CNG-A2 KO and Sema7A KO mice were purchased from The Jackson Laboratory (Nos. 002905 and 005128). BAC Tg mice containing the *MOR29A* and

*MOR29B*[18], Tg mice expressing the *rI7* with the *MOR23* promoter[8], rI7 (RDY)-ires-gap-YFP mice[8], OSN-specific cKO of Nrp1[42], and MOR28-knockin mice[19] were described previously. To generate cell-type-specific Sema7A and PlxnC1 knockout in the Cre driver line, we cloned exon 5 of the *Sema7A* and exon 5 of the *PlxnC1* gene into the double-floxed, inverted open-reading frame plasmid DT-A/ Conditional FW (http://www.cdb.riken.jp/arg/cassette.html). The conditional KOs were generated according to the published protocol (http://www.cdb.riken.jp/arg/ tg.html). The Pcdh21-Cre mouse (RBRC02189) was obtained from RIKEN[24]. (http://www.cdb.riken.jp/arg/tg.html).

All animal experiments were approved by the Animal Care Committee in the University of Fukui and Tokyo and RIKEN institute in Kobe.

**Antibodies**. Antibodies against Sema7A (goat, 1:3000, #AF-1835), PlxnC1 (goat, 1:3000, discontinued), and CNG-A2 (rabbit, 1:200, #APC-045), vGlut2 (guinea pig, 1:1000, #AB2251-l), GluR1 (rabbit, 1:1000, #ab51092), GFP (rabbit, 1:1000, #A-10260), Lucifer yellow (rabbit, 1:2000, #A-5750), and OMP (goat, 1:2000, #Cat. No. 019-22291), were purchased from R&D Systems, Abcam, Alomone Labs, Millipore, Invitrogen, Abcam, Thermo Fisher Scientific, Thermo Fisher Scientific, and Wako Chemicals, respectively. Antibodies against PAK, p-PAK, ERK, and p-ERK (rabbit, 1:1000, #2602, #2606, #4372, #4370) were obtained from Cell Signaling. Antibodies against FLAG (rabbit, 1:1000, #ab2492) and c-myc (mouse, 1:1000, #ab32) were from Abcam.

**In situ hybridization**. To prepare cRNA probes, DNA fragments of 500–3000 bp were amplified by PCR from the OB cDNA of C57BL/6 mice. Only unique sequences were amplified for each gene. PCR products were subcloned into pGEM-T vector (Promega) and used as templates to make cRNA probes. Hybridization was performed according to the standard method[9]. Primer sequences used for PCR are as follows: *sema7A*, 5′-gctccattgcagaaggtttc-3′ (forward) and 5′-gctca-cagctctgttccaca-3′ (reverse); *plxnC1*, 5′-gggactttcaagcgactgag-3′ (forward) and 5′-agtgtcttgcggagatgctt-3′ (reverse); *integrinβ1*, 5′-gaaggtggctttgatgcaat-3′ (forward) and 5′-tgttggaacactttcgtcca-3′ (reverse).

**Immunostaining**. For immunohistochemistry, OB sections were perfused and treated with the primary and secondary antibodies[9]. Sections were photographed with a fluorescence microscope, Model IX70 (Olympus), coupled to a cooled CCD camera, C4742-95-12ERG (Hamamatsu Photonics).

**Immunoelectron microscopy**. Mice were transcardially perfused with 4% paraformaldehyde (PFA) and 0.1% glutaraldehyde (GA) in 0.1 M phosphate buffer (PB), immersed in the same buffer at 4 °C for 1–2 h, and then sectioned transversely into 50-μm slices with a vibratome. The sections were rinsed in 0.1 M PB, incubated in 0.1 M PB with 2% normal serum for 1 h, and then incubated in primary antibody/1% normal serum over-night at 4 °C. The sections were rinsed in 0.1 M PB, incubated with Vector Labs biotinylated secondary immunoglobulin for 60 min, rinsed in 0.1 M PB, incubated in the Molecular Probes Nanogold for 90 min, and again rinsed in 0.1 M PB. The samples were then incubated in 0.1% GA for 10 min, and in Molecular Probes Silver Enhance Kit for 3–5 min. For electron microscopy, the sections were further treated as described in the previous section.

**Naris occlusion**. Unilateral naris occlusion was performed at P0 by electrical cautery and continued to P6. Mice were killed at P6 and OB sections were analyzed by immunohistochemistry for the expression of synaptic markers.

**Intracellular injection**. Intracellular injection[23] was performed as described below. Under a water-immersion ×40 objective lens, a micropipette was inserted into a given glomerulus. After labeling, slices were incubated with a blocking solution containing 5% serum and 0.5% Triton-X in PBS. Samples were incubated with rabbit polyclonal anti-Lucifer yellow antibodies and chicken anti-GFP antibodies, rinsed with PBS, and incubated with Alexa Fluor 488-conjugated goat anti-rabbit IgG (H+L) conjugate and Alexa Fluor 555-conjugated goat anti-chicken IgG (H+L) (Thermo Fisher Scientific, 1:200 dilution). Mature (single dendrite) and immature (multiple dendrites) M/T cells were counted.

**Alkaline-phosphatase (AP) fusion proteins**. To generate fusion proteins, cDNAs corresponding to the ectodomain were introduced into the APtag-5 (GenHunter Corporation) of AP fusion vectors. DNA fragments of the ectodomain were obtained by PCR. This enables the ectodomain to be fused to the N terminus of AP. AP fusion proteins were produced in the transfected HEK293T cells and concentrated using Centriprep (Millipore). Fresh-frozen OB sections were postfixed at −20 °C for 20 min in 100% methanol. AP staining was performed following the standard method[7, 9, 17].

**In vitro binding assay**. Coding sequences of mouse SAP90, PlxnC1, and ectodomain of Sema7A were cloned into pCMV-SC-NF (STRATAGENE), APtag-5 (GenHunter Corporation), and pCMV-SC-NM (STRATAGENE), respectively. FLAG-tag and AP-tag were attached to the N-terminal of SAP90 and Sema7A, respectively. Myc-tag was inserted between the signal peptide and semaphorin domain of PlxnC1. The mutant Sema7A, Y213S, was generated by PCR on the *Sema7A* cDNA (pCMV-AP-secret type Sema7A mut)[17]. The cytosolic domain sequence in the *PlxnC1* gene was deleted by PCR to generate pCMV-c-myc-ΔC PlxnC1. Co-transfection of HEK293 cells was performed as described previously with minor modification[7, 9, 17]. Two days after transfection, HEK cells were fixed on micro slides and processed for immunocytochemistry with antibodies against FLAG, myc, and Sema7A.

**Drug administration**. MK-801 (Sigma–Aldrich) was dissolved in normal saline to achieve a final concentration of 0.02 mg/kg immediately before use. MK801/saline injection was performed with a stainless steel needle (30 gauges), which was directly inserted into the dorsal OB, twice a day from P2 to P5 in two groups of littermates.

**Intensity measurement**. For fluorescent signals of immunostaining, digital images were captured with a digital CCD camera, C4742-95-12ERG (Hamamatsu Photonics), and two-photon microscopy (Olympus). Tone was reversed and a monochrome image was used for the measurement. For staining of the OB sections, digital images were captured with a digital CCD camera, Model DP70 (Olympus). To quantitate the staining level of each glomerulus, the mean pixel intensity within the region surrounded by the periglomerular cell nuclei was measured using Scion Image (Scion Corp.).

**Statistical analyses**. All statistical analyses were performed using Excel 2003 (Microsoft) with the Statcel2 add-on (OMS).

**Data availability**. The data that support the findings in this study are available from the corresponding author on request.

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

## Acknowledgements
We are grateful to H. Sagara for his help in EM analyses. We thank A.J. Otsuka and members of our laboratory for valuable suggestions and discussion. This work was supported by grants from the Ministry of Education, Culture, Sports, Science and Technology of Japan (24000014, 17H06160 to H.S. and 17H05943, 17K19386 to H.Ni.). N.I. was a pre-doctoral fellow of the Japan Society for the Promotion of Science.

## Author contributions
N.I., H.Ni., and H.S. conceived and designed the study. N.I. performed most of the experiments. H.Na. helped dye injection and H.K. generated mutant mice. N.I., H.Ni., and H.S. wrote the paper.

## Additional information

**Competing interests:** The authors declare no competing interests.

