## [Peer Review File · Nature Communications]

Reviewers' comments:

Reviewer #1 (Remarks to the Author):

In the mouse olfactory system, olfactory sensory neurons (OSNs) in the nose each express one of 1000 different types of odorant receptors (ORs). While those with the same OR are dispersed in the nose, their axons converge in a few OR-specific glomeruli in the main olfactory bulb (MOB). Previous studies have revealed molecular mechanisms that help guide OSN axons to selected MOB regions or glomeruli. But whether those axons selectively synapse with the dendrites of certain MOB mitral and tufted (M/T) cells is unknown. In this manuscript, Sakano and colleagues report studies that suggest this does occur and, furthermore, that it involves interactions between *Sema7A* on OSN axons and *PlxnC1* on M/T cell dendrites.

The presented experiments indicate that *Sema7A* is expressed by OSNs in an activity dependent manner during the first week after birth, that *PlxnC1* is expressed by M/T cells, and that KO mice lacking *Sema7A* or *PlxnC1* fail to show markers of synapse formation within OR-specific glomeruli. In addition, the usual maturation of M/T cell dendrites from innervating multiple glomeruli to only one glomerulus is delayed in mutants. Using coexpression of wildtype and mutant molecules in HEK293 cells, the authors provide evidence that *Sema7A* interacts with *PlxnC1* and that this interaction causes interaction of *PlxnC1* with *SAP90*, a postsynaptic density protein. They go on to provide evidence that this interaction leads to *Rac1/Cdc42*-mediated phosphorylation of *PAK*. They further show that the NMDA inhibitor, *MK801*, suppresses dendrite maturation, suggesting that synaptic transmission triggers primary dendrite selection by M/T cells.

This is an interesting set of studies that provides new information on how the olfactory system becomes wired up during development. However, I suggest that the following questions/comments be addressed prior to publication.

1. It appears that *Sema7A* is expressed by only some OSNs. If so, what mechanisms would be used by OSNs that lack this protein?
2. "Dendrite maturation" and "dendrite selection" of M/T cells should be defined and explained. Otherwise, it is incomprehensible.
3. In many places, it would be better to state that the results 'suggest' rather than 'indicate' (e.g. p.8 last sentence of first paragraph). Also see last sentence of results section.
4. Fig. 2b. Need better quality photos showing *PlxnC1* is M/T cell dendrites.

Reviewer #2 (Remarks to the Author):

The development of the mammalian olfactory system presents some of the most perplexing problems in developmental neurobiology. While much attention has been paid to the convergent projection of the olfactory sensory neurons (OSNs) to their target glomeruli, the formation of synaptic connection between the OSNs and their synaptic targets is less understood. In this study, Inoue and colleagues investigated the role played by *Sema7A* and its cognate receptor *PlxnC1* in the development of OSN-MC synapse. They show that the two proteins are expressed in separate synaptic partners and their expression is developmentally regulated. The authors performed a comprehensive array of experiments, which provide evident leading to a model in which *Sema7A* in the OSNs regulates synapse maturation through interaction with *PlxnC1* on the mitral cell dendrites. In this model, *Sema7A* expression in the sensory neurons is regulated by odor-evoked activity, and it promotes synapse maturation through *PlxnC1* mediated signaling pathway. This model is intriguing and presents a potential mechanism by which neural activity regulate synapse

formation in the olfactory system. However, I find it difficult to assess the evidence supporting the model. In many cases, primary results are not presented, making difficult to judge whether the conclusion is adequately supported. While the authors investigated from multiple perspectives, each set of conclusions demand more attention (detailed below) as most were based on histological analyses with scant support from other approaches. The most important conclusion, that interfering with *Sema7A/PlxnC1* signaling causes deficiencies in synapse formation, needs better support.

Major concern:

1. The images presented in the manuscript are generally not of high quality and are often color saturated, making it hard to see detailed structures. The images are also over-cropped. While focused examples of staining in synapses, neurons and glomeruli are necessary, larger structures should be shown to provide a perspective of how representative the cropped images are. In keeping with the convention of the fields, please present images of the OE to include the basal lamina and sustentacular cell layers, and those of OB to include IPL, MC, EPL, GL and ONL structures.
2. The specificity of *Sema7A* and *PlxnC1* expression is only assessed by immunostaining and in situ hybridization. These results are not consistent with publicly available data. For example, from the BAC transgene database Gensat.org, *Sema7A* nor *PlxnC1* is specific to sensory neuron. The authors should address the discrepancy.
3. The in situ hybridization images shown in Figure 1D are not consistent. The one with r17 seems more sporadic than the one with naris occlusion. Again, the cropping of the images makes it hard to assess.
4. The EM picture shown in Figure 1b should include larger scale images. It is difficult to know whether *Sema7A* KO alters the synaptic structures. The glomeruli contain several cell types and different synaptic structures. A detailed analysis is warranted.
5. In Figure 3, the authors performed AP staining in conjunction with antibody staining to demonstrate interactions between the two molecules. The graded levels of *Sema7A*, especially *PlxnC1*, seem to be contradictory to earlier images of *Sema7A* and *PlxnC1* expression. It is puzzling why *PlxnC1* shows different levels of expression here, not earlier. Original images of AP and antibody staining should be included.
6. The authors created and used the *PlxnC1* cKO mouse. It is an important piece of evidence supporting the requirement of *PlxnC1* in synaptogenesis. Since this is the first time this line is used, the authors should include verification that *PlxnC1* is no longer expressed in the *PlxnC1* cKO mice.
7. The authors claim the *Sema7a* knockout specifically affect synaptogenesis but not the OSN axon projection pattern. The conclusion is based on bar graphs of relative density of immunostaining against *VGlut2* and *GluR1*, without the support of original images. It is difficult to assess how well the bar graphs reflect the staining. While *vGlut2* and *GluR1* are important synaptic components, their levels could reflect synaptic plasticity in addition to synaptic structures. Additional analyses, perhaps functional assay, could help convincing the readers. Since the author have performed ultrastructural studies on *Sema7A* KO mice, the EM images may provide additional support of the claim.
8. Judging from the images, the assessment of dendritic maturation seems arbitrary. Ngai and colleagues (Lin et al, 2000) have clearly defined the dendritic morphology of immature, transitory and mature mitral cells. The images shown in Figure 4 and 7 do not contain examples of the different types for each genotype. There seems to be no image showing a single mitral cell innervating two or more glomeruli. Thus, it is not clear how the percentage of immature mitral cells are calculated. It is curious as to why injection into the glomeruli only result in labeling a single cell in the examples presented. I would think that the method is likely to result in labeling of multiple cells.
9. Table S1 described the quantification of some properties of glomerular formation. The data is difficult to assess without original images.

Minor concern:

1. Figure 1. Please check scale bars. It appears the scale bar in (a), (c) and (d) are inconsistent

with the known scales of OSN and glomeruli. The shape and position of the circles indicating the glomeruli are not consistent. In (d), there is no ONL in the OE.

2. Fig1d right panel. DRY –mutant. I think the authors should refer the wt as (DRY) if the mutant is labeled as RDY.

3. Figure 2. Labeling of GL and EPL are not consistent. It is also important to show the mitral cell layers. Are the cell bodies stained for PlxnC1? The PlxnC1 staining results at 8w is not consistent with the bar graph.

4. Gene and protein symbols should be consistently used. Sometime different names were used, for example, CNG-A2.

5. It is not clear how the authors assess whether staining in the glomeruli is from OSN or MC in Table 1.

6. It is not clear which part of the bulb and glomeruli were sample to quantify gene expression level. Are control and mutant matched in their position for quantification purpose? The details should be provided.

7. The method description is too brief. Some of the critical steps are missing. Abbreviations such as PB, GA are not explained.

8. The JAX catalog number for the mouse strains should be identified.

9. The catalog number and possibly the lot number for the antibodies used should be included.

10. In situ probe sequence should be provided.

11. I hope the discussion could be more circumspective and less speculative.

Reviewer #3 (Remarks to the Author):

In the manuscript "Sema7A/PlxnCI signaling is essential for triggering the activity-dependent olfactory synapse formation" Sakano et al tackle a very important problem namely how synapses between ORNs and M/TCs are formed, as an example of circuit specific synapse formation. They demonstrate the involvement of 2 key players, Sem7A in OSNs and PlxnC1 in MTC dendrites. They use a combination of various (established and newly generated) transgenic mousselines, IHC, AB stains, morphological and ultrastuctural analysis and biochemical investigations.

This is an exceptionally thorough and comprehensive study that "bails" the Sem7A/PlxnC1 interaction as a key mechanism inducing synapse formation.

I have only small comments to possible further enhance the clarity of the work:

(1) Overall the manuscript is written in a very condensed form. More expanded methods, further explanations in the text (especially in the descriptions of Fig 3-6) and maybe even the presentation of more, intermediate data (e.g. examples of the Sem7A-AP stains underlying the aggregate data in Fig3) are in my opinion essential to reach the wide audience of Nat Comm

(2) Figure 4b,d : Please give absolute numbers for the number of glomeruli investigated and the average number of M/T cells (means+-SD) and the average fraction of mature or immature M/T cells (means SD). I don't understand what "12 pairs" means. Same for Fig 5 right.

(3) I don't understand how Figure 6b is quantified for 6c – please expand

(4) Page 8, line 2 , should be Fig 6b, and c (not Fig 4)

(5) The authors use several OR-IRES-SEM7A-IRES-YFP mouse lines. It had been reported previously that double IRES constructs can result in perturbed glomerulus formation. Can the authors comment whether they observe e.g. smaller glomeruli etc? To be clear – this is not relevant to the authors results but would be an interesting observation.

We are pleased to find that overall comments from all three reviewers are quite positive and would like to thank them for their constructive criticisms and suggestions. We really appreciate their time and effort in reading our manuscript and raising various points. We acknowledge that these comments were quite helpful in improving our paper. Although the time for revision was limited, we made every our effort to respond to the reviewers' comments and to address their questions.

Point-by-point responses to each reviewer are as follows (reviewer's comments are in bold letters):

Reviewer #1 (Remarks to the Author):

1. It appears that Sema7A is expressed by only some OSNs. If so, what mechanisms would be used by OSNs that lack this protein?

Although expression levels of Sema7A vary among glomeruli, all OR species generate basal levels of Sema7A probably due to the spontaneous neuronal activity of OSNs (Reisert, J., *J. Gen. Physiol.* **136**. 529-540. 2010). Since Sema7A expression was totally abolished by the CNG-A2 KO or by RDY mutation (Fig. 1d), we assume that the activity is generated by OR molecules via CNG channels. These are now mentioned in page 5, line 6-9.

2. "Dendrite maturation" and "dendrite selection" of M/T cells should be defined and explained. Otherwise, it is incomprehensible.

We examined the maturity of M/T-cell dendrites by analyzing whether the primary dendrite has been selected. This is now stated in the text, page 7, line 1-2 and also in the legend to Fig.4. To avoid confusion, we changed "dendrite maturation" to "dendrite selection" throughout the text wherever appropriate (page 6, line21-22; page 7, line 3, 9, 11, 17, 19; page 9, line 16, 18; page 11, line 8, 22, 24).

3. In many places, it would be better to state that the results 'suggest' rather than 'indicate' (e.g. p.8 last sentence of first paragraph). Also see last sentence of results

section.

According to the reviewer's suggestion, we changed "indicate" to "suggest" in the text (page 7, line 16; page 8, line 19; page 9, line 12, page 10, line 2).

4. Fig. 2b. Need better quality photos showing PlxnC1 in M/T cell dendrites.

Due to the quality of antibodies, staining photos had some background in Fig. 2b. To improve the figure, we replaced old pictures with new ones covering larger areas of M/T-cell dendrites, so that localization of staining signals can be seen more easily.

Reviewer #2 (Remarks to the Author):

Major concern:

1. The images presented in the manuscript are generally not of high quality and are often color saturated, making it hard to see detailed structures. The images are also over-cropped. While focused examples of staining in synapses, neurons and glomeruli are necessary, larger structures should be shown to provide a perspective of how representative the cropped images are. In keeping with the convention of the fields, please present images of the OE to include the basal lamina and sustentacular cell layers, and those of OB to include IPL, MC, EPL, GL and ONL structures.

Responding to the reviewer's comments, we made every effort to improve the figures. As suggested, low magnification photos covering larger areas are now shown, so that the readers can tell where the cropped pictures were located. In Fig. 1d, we added new images covering the basal lamina and sustentacular cell layer for the OE, and the ONL, GL, EPL, MC and IPL for the OB. In Fig. 2b, a large scale photo is now shown covering the ONL, GL, EPL and MC. In Fig. 4a and c, we added new photos covering the ONL, GL and EPL. In Supplementary figure 1, new figures were added covering

the GL, EPL and MCL. Also for other figures, we tried to replace old photos with new ones of high quality and cover layer areas to avoid over-cropping.

2. The specificity of Sema7A and PlxnC1 expression is only assessed by immunostaining and in situ hybridization. These results are not consistent with publicly available data. For example, from the BAC transgene database Gensat.org, Sema7A nor PlxnC1 is specific to sensory neuron. The authors should address the discrepancy.

As pointed out, Sema7A and PlxnC1 were reported to be expressed in other tissues. However, they were detected only by *in situ* hybridization in most cases and expression levels were quite low. In our study, Sema7A expression in OSN axons detected by *in situ* hybridization was also verified by immunostaining and immune-electron microscopy. It should be noted that Sema7A was not detected in other types of cells surrounding the glomerular structure. PlxnC1 was also detected in M/T cell dendrites by immunostaining in our study and was not found in the surrounding cells outside of the glomeruli. In the M/T-cell-specific conditional KO (cKO), PlxnC1 expression was totally abolished within the glomerular structure. Furthermore, no sign of PlxnC1 expression was found in the surrounding cells in the cKO. This is now mentioned in the text, page 7, line 5-6, and shown in new Supplementary figure 1.

3. The in situ hybridization images shown in Figure 1D are not consistent. The one with rI7 seems more sporadic than the one with naris occlusion. Again, the cropping of the images makes it hard to assess.

As pointed out, old Fig. 1d was not convincing and its resolution was poor. We therefore replaced it with a new photo with better resolution, where *in situ* hybridization signals are clearly separated. As for the RDY mutant study, it should be mentioned that OR signals were blocked only in the rI7 OSNs because the DRY→RDY mutation was introduced only in rI7, but not in other OR species expressed in the surrounding glomeruli.

4. The EM picture shown in Figure 1b should include larger scale images. It is difficult to know whether Sema7A KO alters the synaptic structures. The glomeruli contain several cell types and different synaptic structures. A detailed analysis is warranted.

As suggested, we analyzed synaptic contacts and post-synaptic structures by electron microscopy (EM) in both the WT, and Sema7A total KO. Synaptic structures, e.g., post-synaptic density (PSD), were rarely found in the KO at P5, although synaptic contacts were detected within glomeruli (see the attached photo below). This is now mentioned in the text, page 7, line 12-14. Since we plan to publish these new EM data in a separate paper, these photos are not included in the present paper.

Post synaptic structure

EM analysis of post-synaptic structures. Synaptic contacts and post-synaptic structures were studied in the WT, and Sema7A total KO by EM. Arrows indicate synaptic contacts. Enlarged views demonstrate that only the surface attachment, but not mature PSD, is formed in the KO. *** $p < 0.005$ (Student's t-test). n.s., not significant. Error bars indicate SD (n=3, 4 animals). Scale bars, 300 nm.

5. In Figure 3, the authors performed AP staining in conjunction with antibody staining to demonstrate interactions between the two molecules. The graded levels of Sema7A, especially PlxnC1, seem to be contradictory to earlier images of Sema7A and PlxnC1 expression. It is puzzling why PlxnC1 shows different levels of expression here, not earlier. Original images of AP and antibody staining should be included.

As pointed out, old Fig. 3 was confusing because the scales of x and y axes were different in the left and right figures. We now replotted the data in the same scale in the new figure. As shown in the new plot, expression levels of Sema7A in glomeruli are variable, whereas those of PlxnC1 are almost even. These new plots are not contradictory to the earlier images of Sema7A and PlxnC1 in Fig. 1 and 2. We also added photos of AP-staining in Fig. 3.

6. The authors created and used the PlxnC1 cKO mouse. It is an important piece of evidence supporting the requirement of PlxnC1 in synaptogenesis. Since this is the first time this line is used, the authors should include verification that PlxnC1 is no longer expressed in the PlxnC1 cKO mice.

As suggested, we added detailed description of the PlxnC1 cKO in the text, page 7, line 5-8. In the Supplementary figure 2, we added new data showing that PlxnC1 expression is indeed abolished in the OB of the cKO. We also studied synaptic structures by EM. In the PlxnC1 cKO, synaptic contact was seen, however, PSD formation was blocked as in the Sema7A KO. This is now mentioned in the text, page 7, line 12-14.

7. The authors claim the Sema7a knockout specifically affect synaptogenesis but not the OSN axon projection pattern. The conclusion is based on bar graphs of relative density of immunostaining against VGlut2 and GluR1, without the support of original images. It is difficult to assess how well the bar graphs reflect the staining. While vGlut2 and GluR1 are important synaptic components, their levels could reflect synaptic plasticity in addition to synaptic structures. Additional analyses, perhaps functional assay, could help convincing the readers. Since the author have performed ultrastructural studies on Sema7A KO mice, the EM images may provide additional support of the claim.

Responding to the reviewer's comment, we added typical images of immunostaining of vGlut2 and GluR1 in Fig. 4a and c. The reviewer also pointed out the possibility of synaptic plasticity that may affect the levels of synaptic markers. As suggested, we

performed EM studies on the synapse formation in the Sema7A KO and PlxnC1 cKO. We found that physical contacts were formed between the OSN axons and M/T-cell dendrites, however, PSD formation and synapse maturation were perturbed (please see the attached photos in response 4). We therefore assume that synapse formation itself, but not the plasticity, is affected in the KO mice. These observations are now mentioned in the text, page 7, line 12-14.

8. Judging from the images, the assessment of dendritic maturation seems arbitrary. Ngai and colleagues (Lin et al, 2000) have clearly defined the dendritic morphology of immature, transitory and mature mitral cells. The images shown in Figure 4 and 7 do not contain examples of the different types for each genotype. There seems to be no image showing a single mitral cell innervating two or more glomeruli. Thus, it is not clear how the percentage of immature mitral cells are calculated. It is curious as to why injection into the glomeruli only result in labeling a single cell in the examples presented. I would think that the method is likely to result in labeling of multiple cells.

First, it should be mentioned that our criteria of M/T-cell maturation is different from those in the paper of Dr. Ngai's group. In our experiment, we injected Lucifer yellow (LY) into the glomerular structure. Therefore, only M/T cell dendrites connecting to the glomeruli were detected by LY. We counted immature M/T cells with multiple dendrites connecting to the glomeruli. In the experiments reported by Dr. Ngai and colleagues, DiI was injected into the M/T cell axons and stained dendrites even when they did not reach the glomeruli. Thus, more numbers of immature M/T cells were detected in their DiI injection. We counted LY-stained M/T-cells with a primary dendrite as mature and those with multiple dendrites as immature.

For single-cell staining, detailed injection procedures are now described in the Method section referring the original paper (Naritsuka et al., *J. Comp. Neurol.*, 515, 409, 2009). Unlike DiI that diffuses in the cell membrane, LY is a low molecular-weight dye that is water-soluble. In our injection, other types of cells than M/T cells, e.g., olfactory sensory neurons and periglomerular cells, are sometimes stained. However, we confirmed that single M/T cells were indeed stained by LY injection into a specific

glomerulus, and then the samples processed for further studies. Thus, our LY injection allows us to visualize single M/T cells connecting to a particular glomerulus.

9. Table S1 described the quantification of some properties of glomerular formation. The data is difficult to assess without original images.

Responding to the reviewer's comment, we added two sets of whole-mount OB views for each transgenic animal in Supplementary figure 1. One set shows the glomerular locations of MOR29A in the WT, Sema7A KO, and Nrp1 cKO. The other set is for the MOR28 glomeruli in the WT and Sema7A KO. Statistical analyses were performed based on these original images and are now shown in Supplementary table 1.

Minor concern:

1. Figure 1. Please check scale bars. It appears the scale bar in (a), (c) and (d) are inconsistent with the known scales of OSN and glomeruli. The shape and position of the circles indicating the glomeruli are not consistent. In (d), there is no ONL in the OE.

Corrections were made for scale bars in new Fig. 1a, c, and d. Dotted circles indicating glomerular structures were carefully rewritten tracing the periglomerular cells. The ONL in Fig. 1d is connected to OSN.

2. Fig1d right panel. DRY –mutant. I think the authors should refer the wt as (DRY) if the mutant is labeled as RDY.

As advised, we referred WT as WT (DRY) in Fig. 1d.

3. Figure 2. Labeling of GL and EPL are not consistent. It is also important to show the mitral cell layers. Are the cell bodies stained for PlxnC1? The PlxnC1 staining results at 8w is not consistent with the bar graph.

As suggested, we added low magnification photos covering the mitral cell layer.

PlxnC1 was found localized in the dendrites of M/T cells, but not in the cell bodies. The last point raised by the reviewer that “The PlxnC1 staining results at 8W in not consistent with the bar graph” may be misunderstanding. It is possible that “Integrin β 1” was confused with “PlxnC1”. Integrin β 1 was detected in M/T-cell bodies at 8w.

4. Gene and protein symbols should be consistently used. Sometime different names were used, for example, CNG-A2.

As advised, we made necessary changes for CNG-A2 in page 4, line1 and 8 and in Fig. 1d.

5. It is not clear how the authors assess whether staining in the glomeruli is from OSN or MC in Table 1.

As the reviewer mentions, it was not easy to assess whether staining in the glomeruli was from OSN axons or M/T-cell dendrites. However, this screening was to search for the candidate genes using *in situ* hybridization, asking what kind of genes were expressed in OSNs and M/T cells (Table 1). After this screening, we closely looked at the localization of candidate molecules in M/T-cells dendrites by immunostaining. Among them, PlxnC1 and Pcdh21 were found to be detected in the dendrites of M/T cells, but not ephrin-A5, Kirrel3, Ptpn, Nrp1, Nrp2, Sema3A, Sema4d, Sema7A, and PlxnA3 (Table 1).

6. It is not clear which part of the bulb and glomeruli were sample to quantify gene expression level. Are control and mutant matched in their position for quantification purpose? The details should be provided.

When we quantify the levels of Sema7A, PlxnC1, vGlut2, and GluR1 in the OB, we focused on specific glomeruli, e.g., for r17, MOR29A, or MOR29B. This is mentioned in the legends to Fig. 1c, 1d, 4a, 4b, 5, and 7a-c. No notable differences were found between the WT and mutant mice in the axonal projection of OSNs, sizes of OBs, and

glomerular positions for some specific OR species (Supplementary table 1).

7. The method description is too brief. Some of the critical steps are missing. Abbreviations such as PB, GA are not explained.

As advised, detailed procedures were added in the Methods section, e.g., for immunostaining in page 14,15 line 17-24, 1-2; *in situ* hybridization, in page 15, line 9-12; immunoelectron microscopy, in page 15, line 21-22; and naris occlusion, in page 16, line 10. Abbreviations are now explained in page 15, line 21-22.

8. The JAX catalog number for the mouse strains should be identified.

Catalog numbers are now included for CNG-A2 (002905) and Sema7A KO (005128) in page 14, line 3.

9. The catalog number and possibly the lot number for the antibodies used should be included.

Catalog numbers for all antibodies used in our present study are now included in the Methods section in page 14,15 line 17-24, 1-2.

10. In situ probe sequence should be provided.

As advised, primer sequences used to amplify a part of cDNA is now described for *Sema7A*, *PlxnC1*, and *Integrin β 1* genes in page 15, line 9-12. The amplified cDNAs were used to generate cRNA probes for *in situ* hybridization.

11. I hope the discussion could be more circumspective and less speculative.

As advised, speculative discussions were removed or rephrased, e.g., in page 11, line 6-7, 14-15; page 11, line 9-10; and page 12, line 5-6; and page 13, line 12-21.

Reviewer #3 (Remarks to the Author):

1. Overall the manuscript is written in a very condensed form. More expanded methods, further explanations in the text (especially in the descriptions of Fig 3-6) and maybe even the presentation of more, intermediate data (e.g. examples of the Sem7A-AP stains underlying the aggregate data in Fig3) are in my opinion essential to reach the wide audience of Nat Comm.

Responding to the reviewers' comments, we tried our best in improving the paper. For specific points, we made necessary changes and added more information as follows: As advised, more explanations for the data in Fig. 3 were added in page 5, lines 23 to page 6, line 8. In Fig. 4a and c, staining photos were added for quantification of synapse markers. Further explanations of methods were added in page 7, line 20-21 for Fig. 5, and page 8, line 9-11 for Fig. 6. More detailed procedures were described in the Methods section in page 14,15 line 17-24, 1-2; page 15, line 9-12; page 15, line 21-22; and page 16, line 10. Abbreviations are now explained in page 15, line 21-22.

2. Figure 4b,d : Please give absolute numbers for the number of glomeruli investigated and the average number of M/T cells (means+-SD) and the average fraction of mature or immature M/T cells (means SD). I don't understand what "12 pairs" means. Same for Fig. 5 right.

We agree that the numbers described in the text were not well explained. In the manuscript, all n's are the numbers of glomeruli.

In Fig. 4b, twelve mice (n=12) were analyzed each for the WT and Sema7A KO. A small amount of LY was carefully injected into the glomerular structure to label a limited number of M/T cells. As a result, 20 glomeruli for each were successfully stained and analyzed further for the maturity of connecting M/T-cell dendrites. The maturity was determined by analyzing whether the primary dendrite had been selected: mature, with a primary dendrite; and immature, with multiple or branched dendrites. In the WT, 76.5% (26/34) of stained M/T cells were mature, whereas in the KO, the maturity was 23.5% (8/34).

In Fig. 4d, fourteen mice were analyzed each for the WT and PlxnC1 cKO. Among the LY-injected glomeruli, 23 of them contained a few well-stained M/T cells. In the WT, 77.4% (24/31) and in the cKO, 22.6% (7/31) of the stained M/T cells possessed a primary dendrite.

In Fig. 5, eight Tg-MOR29A mice were analyzed for the maturity of M/T-cell dendrites in the MOR29A and MOR19B glomeruli. Here, sixteen glomeruli for each were nicely stained and analyzed further. In the MOR29A glomeruli, 27.6% (8/29) of stained M/T cells possessed a primary dendrite, and in the MOR29B glomeruli, 78.6% (22/28) had a selected dendrite.

More explanations were added in Fig.7. For the maturity of M/T-cell dendrites, nineteen glomeruli were analyzed in the MK801 experiment. For the MK801⁻ glomeruli, 19/25 (76.0 %) of stained M/T cells possessed a primary dendrite, whereas for the MK801⁺ glomeruli, the maturity was much lower, 6/25 (24.0 %).

These are now mentioned in the legends to Figs. 4, 5, and 7.

3. I don't understand how `Figure 6b is quantified for 6c – please expand

In the immunostaining in Fig. 6b, FLAG-tagged SAP90 and myc-tagged PlxnC1 are detected in red and green, respectively. We counted the numbers of yellow (double positive) and green (single positive) dots for PlxnC1 in each HEK293 cell. When no interaction is taking place between PlxnC1 and SAP90, green and red signals are mostly separated. In contrast, yellow cells can frequently be seen, when SAP90 is recruited to PlxnC1.

In the figure, ratios of yellow / yellow and green signals (co-localizing PlxnC1 with SAP90/ total PlxnC1) are shown. This is now explained in the legend to Fig. 6c.

4. Page 8, line 2 , should be Fig 6b, and c (not Fig 4)

These are now corrected as pointed out.

5. The authors use several OR-IRES-SEM7A-IRES-YFP mouse lines. It had been reported previously that double IRES constructs can result in perturbed

glomerulus formation. Can the authors comment whether they observe e.g. smaller glomeruli etc? To be clear – this is not relevant to the authors results but would be an interesting observation.

As pointed out, the numbers of cells expressing the Tg construct were indeed reduced and expression levels of the *Tg Sema7A* flanked by *IRES* sequences were lowered. Since, this double IRES construct was not used in the experiments in the present paper, we deleted the description of this construct.

REVIEWERS' COMMENTS:

Reviewer #1 (Remarks to the Author):

The authors have addressed all my concerns.

Reviewer #2 (Remarks to the Author):

Through a series of well designed experiments, the authors demonstrate the requirement of *Sema7A/PlxnC1* signaling in OSN/mitral cell synapse formation and maturation. They also demonstrate the regulation of this signaling pathway through neural activity in the OSNs, and provide evidence of a signaling cascade linking *Sema7A* with synapse formation. The authors have addressed most of the concerns I had over the previous submission.

A minor inconsistency is that the legend for Figure 4d states the data is from P6 when the text and figure label it at P5.

Reviewer #3 (Remarks to the Author):

The authors addressed all my (anyway minor) comments very well and I support publication in Nat Comm. Congratulations again on a very important contribution.